# Evidence on strategies promoting discharge planning participation in patients with chronic heart failure: A scoping review protocol

Yanyan Fan[1☯], Haotian Zhang[2☯], Enyuan Yan[2], Haoxin Tian[2], Yuntao Yu[2], Shouqing Lin[ID][3]*

**1** School of Nursing, Chinese Academy of Medical Sciences & Peking Union Medical College, Beijing, China, **2** School of Nursing, Binzhou Medical University, Yantai, Shandong, China, **3** Department of Obstetrics and Gynecology, Peking Union Medical College Hospital, Chinese Academy of Medical Sciences & Peking Union Medical College, Beijing, China

☯ These authors are contributed equally to this work.
* y51617@126.com

## Abstract

### Introduction

Chronic heart failure (CHF) poses a significant global healthcare burden. High readmission rates, driven by inadequate discharge preparation and suboptimal transitional care, underscore the need for effective discharge planning. Patient participation in discharge planning is essential for optimal outcomes, yet evidence on strategies promoting such participation remains fragmented.

### Aim

To systematically identify, map, and synthesize evidence on strategies promoting patient participation in discharge planning for adults with CHF, and to map the range of outcomes reported in relation to such strategies.

### Methods

Following JBI methodology and PRISMA-ScR guidelines, we will search 20 databases (including PubMed, CINAHL, Embase, Cochrane Library, Web of Science, Chinese databases) from inception. Studies of any design addressing patient participation in discharge planning for adults (≥18 years) with CHF will be included. Two reviewers will independently screen studies and extract data on study characteristics, participation strategies, and reported outcomes. Data will be synthesized using a thematic synthesis methodology, with findings presented in tables, figures, and narrative format; the PAGER framework will guide the identification of patterns, advances, gaps, and evidence pertinent to clinical practice.

**Data availability statement:** Deidentified research data will be made publicly available when the study is completed and published.

**Funding:** The author(s) received no specific funding for this work.

**Competing interests:** The authors have declared that no competing interests exist.

## Conclusion

This scoping review will systematically map evidence on strategies promoting patient participation in discharge planning for adults with CHF and describe reported outcomes. By identifying effective approaches and evidence-practice gaps, this review will inform clinical practice, policy development, and future research to enhance patient-centered discharge planning for this vulnerable population. Results will be disseminated via peer-reviewed publication and stakeholder engagement.

## Introduction

Chronic heart failure (CHF) is a life-threatening syndrome with substantial morbidity and mortality [1]. Epidemiological data reveal that the prevalence rate of heart failure in adults worldwide ranges from 1% to 3% [2]. The burden of CHF extends beyond its high morbidity and mortality rates to encompass substantial healthcare resource utilization and economic costs. Patients with CHF experience frequent hospitalizations, with 40.5% of hospitalized cases having three or more admissions [3]. The readmission burden is particularly concerning, with 18.2% of patients requiring readmission within 30 days and 31.2% within 90 days [4]. These recurrent hospitalizations contribute to an estimated annual per-capita healthcare expenditure of $4,406.80, imposing a significant economic burden on healthcare systems [3].

Multiple factors contribute to the persistently high readmission rates among CHF patients. Research has identified that inadequate discharge planning, poor care transitions, non-adherence to medication regimens, insufficient patient education, lack of patient participation in the discharge process, and inadequate post-discharge follow-up are significant contributors to preventable readmissions [5,6]. These findings underscore the critical need for effective discharge planning as a cornerstone intervention to improve care transitions and reduce readmissions.

In response to this challenge, healthcare policy has increasingly emphasized the importance of structured discharge planning and transitional care. The Shandong province government in China has launched the Inpatient Discharge Planning Project to improve transitional care for chronic disease patients [7]. The American Heart Association (AHA) guidelines explicitly recommend that post-discharge systems of care be used to facilitate transitions to effective outpatient care for patients hospitalized with heart failure [5]. These guidelines emphasize comprehensive discharge planning, including patient education, medication reconciliation, early follow-up within 7–14 days, and multidisciplinary disease-management programs for high-risk patients [5,8].

Evidence demonstrates that patient-centered discharge planning can significantly reduce hospital length of stay, decrease readmission rates, and enhance patient satisfaction [9]. Furthermore, the effectiveness of discharge planning is substantially enhanced when patients are actively engaged as partners in the process, rather than serving as passive recipients of care instructions [10,11]. Patient participation is a multidimensional concept central to contemporary nursing practice. Cahill identified

its core attributes through concept analysis, emphasizing the establishment of a nurse–patient relationship, the narrowing of information and knowledge gaps, a degree of power surrendered by the nurse, and active patient participation with positive benefit [12]. Sahlsten et al. similarly defined patient participation as comprising an established relationship, power-sharing, shared information, and active joint participation, with nurses assuming roles as empowerers and supportive facilitators [13]. More broadly, Paukkonen et al. emphasized that patient participation enables patients to get involved in and affect their own care in partnership with healthcare providers—a dynamic particularly important in populations with multimorbidity, as individualized participation is recognized as essential for meeting complex care challenges [14]. This concept has gained increasing recognition in healthcare policy and practice, exemplified by legislative reforms such as the revised Swedish Patient Act, which explicitly mandates patient participation to create a more balanced power dynamic between patients and healthcare providers [15]. It is important to note that the terms patient participation, patient involvement, and patient engagement are often used interchangeably in the literature [13,14]. Jerofke-Owen et al. concluded that, from a nursing perspective, all three describe a patient playing an active role in their healthcare, sharing the same core of establishing a mutual nurse–patient partnership toward empowerment. This review therefore treats them as conceptually equivalent and includes studies using any of these terms to describe patients' active role in discharge planning [16].

However, its implementation in discharge planning has not been sufficiently improved in practice [5,17]. A qualitative metasummary and an observational study indicate that common barriers include healthcare professionals' failure to elicit or encourage patients' participation preferences, discharge decisions made unilaterally by clinicians without meaningful consultation with patients or families [18], inadequate patient education about discharge options [19]. These barriers may be particularly pronounced among patients with CHF, who face unique challenges that can hinder their participation in discharge planning.

Patients with CHF face unique challenges that may hinder their participation in discharge planning. The complexity of CHF management requires patients to navigate multiple medications, dietary restrictions, fluid management, and symptom monitoring [20]. Additionally, physical symptoms such as fatigue and dyspnea may also limit patients' capacity to participate in lengthy discharge planning sessions [21]. Furthermore, cognitive impairment, which affects CHF patients, further complicates their ability to participate meaningfully in discharge discussions and retain critical self-care information [22]. Notably, the multimorbidity burden common in this population adds layers of complexity to discharge planning, which necessitates tailored, individualized participation strategies [23]. Given these multifaceted challenges, identifying effective strategies to promote meaningful patient participation in discharge planning is essential for this vulnerable population.

Some strategies have been proposed to promote patient participation in discharge planning for adults with CHF, such as provision of easy-to-understand information [17], participation in decision-making processes [17], coordination with multidisciplinary teams [24], use of teach-back methods [25], and shared goal-setting [26]. However, the evidence base remains fragmented across diverse sources and study designs. Additionally, no scoping review has specifically focused on synthesizing strategies to promote patient participation in discharge planning for CHF patients and describing their reported outcomes.

### Review objectives

1. To identify and map strategies and interventions implemented to enhance patient participation in discharge planning for adults with CHF.

2. To identify and describe the range of outcomes (clinical, patient-reported, and healthcare utilization) measured or reported in studies of patient participation in discharge planning for adults with CHF.

3. To identify evidence-practice gaps and propose implications for clinical practice, policy, and future research.

## Methods

### Design

This scoping review will be conducted following the methodological framework proposed by Arksey and O'Malley [27] and further refined by Levac et al. [28], as well as the updated methodological guidance from the Joanna Briggs Institute (JBI) for scoping reviews [29]. The reporting will adhere to the Preferred Reporting Items for Systematic Reviews and Meta-Analysis Protocols (PRISMA-P) (S1 Appendix) and the PRISMA Extension for Scoping Reviews (ScR) guidelines (S2 Appendix) [30,31]. Furthermore, we will employ the PAGER framework (Patterns, Advances, Gaps, Evidence for Practice, and Research recommendations) to guide data charting and the structured presentation of findings, complementing the PRISMA-ScR reporting requirements [32]. Thematic data synthesis will follow the three-stage framework described by Thomas and Harden to systematically identify and interpret key patterns across the included evidence [33]. The protocol is registered with Open Science Framework (https://doi.org/10.17605/OSF.IO/QPAZM). No personal or identifying data will be collected during this scoping review. As such, research ethics board approval is not required.

### Stage 1: Identifying the research question

1. What strategies or interventions have been implemented to promote patient participation in discharge planning for adults with CHF?

2. What outcomes (clinical, patient-reported, and healthcare utilization) have been measured or reported in studies of patient participation in discharge planning for adults with CHF?

3. What evidence-practice gaps exist, and what are the implications for clinical practice, policy development, and future research?

### Stage 2: Identifying relevant studies

**Search strategy.** In collaboration with a health sciences librarian, the following 20 databases and websites will be searched from inception: BMJ Best Practice, Joanna Briggs Institute (JBI), UpToDate, Scottish Intercollegiate Guidelines Network (SIGN), National Institute for Health and Care Excellence (NICE), National Guideline Clearinghouse (NGC), Registered Nurses' Association of Ontario (RNAO) database, Guidelines International Network (GIN), Heart Failure Society of America (HFSA), Medlive, MedSci, PubMed, Web of Science, Cochrane Library, CINAHL, Embase, Chinese Biomedical Literature Database (SinoMed), WANFANG DATA, China National Knowledge Infrastructure (CNKI) database and CQVIP. Reference lists of included studies will be manually searched in addition to the above databases.

The search strategy developed for PubMed (**Table 1**) will be adapted for other databases with appropriate modifications to database-specific syntax and controlled vocabulary.

### Eligibility criteria

Following JBI guidelines, eligibility will be established using the PCC framework:

**Inclusion criteria.** Population: Adults (≥18 years) diagnosed with CHF (any etiology, NYHA class I–IV, or ejection fraction type).

Concept: Patient participation in discharge planning, defined as patients' active role in discharge-related care processes, encompassing information exchange, goal-setting, empowerment, and collaborative partnership with healthcare professionals. This definition is underpinned by a reciprocal patient–professional relationship in which patients' experiential knowledge and preferences are meaningfully integrated into care. Given the conceptual overlap between participation,

**Table 1. Search strategy in PubMed.**

| DATABASE | MEDLINE |
|---|---|
| PLATFORM | PubMed |
| SEARCH | QUERY |
| #1 | ("heart failure" [MeSH Terms] OR "heart failure" [Title/Abstract] OR "cardiac failure" [Title/Abstract] OR "chronic heart failure" [Title/Abstract] OR "congestive heart failure" [Title/Abstract] OR "cardiac insufficiency" [Title/Abstract] OR "cardiac dysfunction" [Title/Abstract] OR "heart decompensation" [Title/Abstract] OR "myocardial failure" [Title/Abstract]) |
| #2 | ("patient discharge" [MeSH Terms] OR "patient discharge" [Title/Abstract] OR "discharge planning" [Title/Abstract] OR "discharge planning program" [Title/Abstract] OR "patient transfer" [Title/Abstract] OR "post discharge" [Title/Abstract] OR "discharge preparation service" [Title/Abstract] OR "discharge process" [Title/Abstract] OR "discharge management" [Title/Abstract] OR "transitional care"[Title/Abstract] OR "care transition"[Title/Abstract] OR "discharge readiness"[Title/Abstract]) |
| #3 | ("patient participation" [MeSH Terms] OR "patient participation" [Title/Abstract] OR "patient involvement" [Title/Abstract] OR "patient engagement" [Title/Abstract] OR "participation" [Title/Abstract] OR "patient empowerment" [Title/Abstract] OR "willingness to participate" [Title/Abstract] OR "participation willing" [Title/Abstract] OR "desire for participation" [Title/Abstract] OR "shared decision making"[MeSH Terms] OR "shared decision making"[Title/Abstract] OR "patient-centered care"[Title/Abstract] OR "patient activation"[Title/Abstract]) |
| #4 | #1 AND #2 AND #3 |

involvement, and engagement in the literature, studies using any of these constructs to describe patients' active role in the discharge planning process are considered eligible.

Context: Any healthcare setting where discharge planning occurs, including acute hospitals, cardiac units, rehabilitation facilities, transitional care settings, and community or home-based follow-up care as part of the discharge planning continuum.

Evidence source types: Quantitative, qualitative, mixed-methods studies, clinical practice guidelines and evidence summary, as well as relevant reviews (systematic, scoping, or narrative).

**Exclusion criteria.** Studies on acute or pediatric heart failure; studies focusing only on healthcare professionals without addressing patient participation; discharge planning without reference to patient participation; abstracts without full text; editorials, commentaries, or registered protocols without published findings; and non-English or non-Chinese publications.

### Stage 3: Study selection (screening)

Retrieved literature will be uploaded into EndNote and duplicates will be deleted. Two reviewers with evidence-based medicine training will independently conduct literature screening. Titles and abstracts will be read during the preliminary screening. Subsequently, the full texts of the selected citations will be rescreened. Any differences during the screening process will be resolved through discussion with a third expert.

### Stage 4: Charting the data (data extraction)

Data from eligible studies will be extracted using a standardized Microsoft Excel spreadsheet. Two reviewers will independently extract data; discrepancies will be resolved through discussion or arbitration by a third reviewer. Study authors will be contacted if essential data are missing or unclear. The following information will be captured:

1. Study characteristics: Author(s), year of publication, country, study design, sample size, participant characteristics.

2. Strategies/interventions: Description of strategies or interventions to promote patient participation.

3. Outcomes: types of outcomes measured and findings reported in relation to participation strategies.

**Stage 5: Collating, summarizing and reporting the results**

**Phase 1 – Quantitative mapping of study characteristics.** Descriptive statistics will be used to summarize key characteristics including publication year, country, study design, sample characteristics, healthcare setting, types of patient participation strategies, and reported outcome measures.

**Phase 2 – Thematic synthesis of participation strategies and outcomes.** Thematic synthesis will follow framework described by Thomas and Harden [33]. The analysis will proceed through the following steps:

(1) two reviewers will independently code the extracted textual data line-by-line.

(2) related codes will be grouped and developed into descriptive themes that closely reflect the content of the included studies.

(3) descriptive themes will be interpreted and abstracted into analytical themes, examined against the conceptual definition of patient participation—encompassing information exchange, shared decision-making, goal-setting, empowerment, and collaborative partnership—to generate deeper insights beyond the original studies.

Discrepancies between reviewers at any stage will be resolved through discussion or, if necessary, adjudication by a third reviewer. The resulting analytical themes will directly inform the application of the PAGER framework.

**Phase 3 – Application of the PAGER framework.** The findings will be presented and interpreted using the PAGER framework, which includes organizing evidence into Patterns, Advances, Gaps, Evidence for practice, and Research recommendations [32]. The five components and their operational definitions within this review are presented in **Table 2**.

Final findings will be presented as an integrated narrative synthesis accompanied by thematic diagrams, summary tables, and a visual evidence map, in accordance with PRISMA-ScR reporting requirements.

**Stage 6: Consultation**

Key stakeholders, including chronic heart failure patients, caregivers, cardiology nurses, clinicians, and hospital administrators, will be invited to review a summary of preliminary findings and provide feedback on their comprehensiveness and practical implications. Consultations will be conducted in person or via WeChat. All feedback will be anonymously summarized to validate findings, refine evidence-practice gaps, and strengthen recommendations. This stage is not part of the formal data analysis.

**Table 2. Application of the PAGER framework in this scoping review.**

| Code | Component | Operational Definition in This Review |
| --- | --- | --- |
| P | Patterns | Recurring strategy types and outcome domains across studies |
| A | Advances | Novel or emerging approaches beyond standard discharge care |
| G | Gaps | Under-researched populations, settings, or outcomes |
| E | Evidence for Practice | Strategies with sufficient evidence for clinical application |
| R | Research Recommendations | Prioritized areas for future primary research |

## Study status and timeline

This is a prospective protocol and no stages of the review have been completed at the time of submission. The anticipated timeline is as follows: (A) record screening is expected to be completed by May 2026; (B) data extraction is expected to be completed by July 2026 and (C) results are expected to be finalized by September 2026.

## Limitations

This review is expected to have several limitations. Firstly, we will not evaluate the quality of included studies because the primary aim of this scoping review is to comprehensively map interventions that promote patient participation in discharge planning. Future research should conduct rigorous quality appraisals and systematic evaluations of effectiveness before these strategies can be recommended for implementation in clinical practice. Secondly, the review is restricted to papers published in English and Chinese, which means that relevant articles published in other languages will be excluded. Finally, although we aim to capture a broad range of evidence sources, some potentially relevant grey literature or unpublished materials may not be identified.

## Discussion

This protocol outlines the first scoping review to systematically map evidence on patient participation in discharge planning for adults with CHF. While discharge planning for adults with CHF has been extensively studied, no review has specifically synthesized strategies to enhance patient participation and their associated outcomes—a critical gap given the recognized disconnect between policy mandates for participation and actual clinical practice.

## Conclusion

CHF imposes substantial burden through high readmission rates and healthcare costs. While discharge planning is well-established for improving transitions, the critical role of patient participation has not been comprehensively synthesized. This scoping review will systematically map available evidence on strategies promoting patient participation in discharge planning for adults with CHF and describe reported outcomes.

By following rigorous JBI methodology and PRISMA-ScR guidelines, this review will provide a comprehensive evidence map to inform clinical practice, organizational policy, and future research. The findings will address an important knowledge gap with practical implications for improving patient-centered discharge planning for this vulnerable population.

Upon completion, results will be disseminated through peer-reviewed publication and stakeholder engagement to maximize impact. This work aims to advance patient-centered care by ensuring individuals with CHF are meaningfully engaged as partners in their discharge planning, ultimately leading to better outcomes and more efficient healthcare resource use.

## Supporting information

**S1 Appendix. PRISMA-P checklist.**
(PDF)

**S2 Appendix. PRISMA-ScR checklist.**
(PDF)

## Author contributions

**Conceptualization:** Yanyan Fan, Shouqing Lin.

**Formal analysis:** Yanyan Fan, Enyuan Yan.

**Investigation:** Haotian Zhang, Enyuan Yan, Haoxin Tian.

**Methodology:** Yanyan Fan, Haotian Zhang.

**Project administration:** Haotian Zhang.

**Resources:** Yanyan Fan, Enyuan Yan, Haoxin Tian, Yuntao Yu.

**Software:** Yanyan Fan.

**Supervision:** Yanyan Fan, Shouqing Lin.

**Writing – original draft:** Yanyan Fan, Haotian Zhang.

**Writing – review & editing:** Yanyan Fan, Shouqing Lin.

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
