## [Decision Letter · Decision Letter 0]

11 Mar 2026

PONE-D-25-67198Evidence on strategies promoting discharge planning participation in patients with chronic heart failure: A scoping review protocolPLOS One

Dear Dr. Shouqing,

Thank you for submitting your manuscript to PLOS ONE. After careful consideration, we feel that it has merit but does not fully meet PLOS ONE’s publication criteria as it currently stands. Therefore, we invite you to submit a revised version of the manuscript that addresses the points raised during the review process.

We look forward to receiving your revised manuscript.

Kind regards,

Divya Bhandari

Academic Editor

PLOS One

Journal Requirements:

Additional Editor Comments:

Dear Authors,

Thank you for your submission. Your manuscript addresses an important topic, and both reviewers acknowledge the relevance of this work.

I agree with the thoughtful and constructive feedback and suggestions provided by Reviewer 1 and believe their suggestions will help strengthen this protocol. I hope you will also find the input valuable.

Given that this is a study protocol, it is not mandatory to adopt every suggestion where recommendations contradict or may not align with your planned approach. That said, I encourage you to address as many comments as feasible and to consider this revision as an opportunity to further enhance clarity and transparency.

Reviewer 2 notes that the manuscript is presented as a protocol rather than a completed study. As PLOS ONE does publish study protocols, no additional action is required in this regard. However, please ensure that the manuscript clearly adheres to the journal guidelines for protocol submissions and that all required elements are fully detailed.

We look forward to receiving your revised submission and detailed response to the reviewers.

Reviewers' comments:

Reviewer's Responses to Questions

**Comments to the Author**

1. Does the manuscript provide a valid rationale for the proposed study, with clearly identified and justified research questions?

Reviewer #1: Yes

Reviewer #2: No

2. Is the protocol technically sound and planned in a manner that will lead to a meaningful outcome and allow testing the stated hypotheses?

Reviewer #1: Yes

Reviewer #2: Partly

3. Is the methodology feasible and described in sufficient detail to allow the work to be replicable?

Reviewer #1: No

Reviewer #2: No

4. Have the authors described where all data underlying the findings will be made available when the study is complete?

Reviewer #1: Yes

Reviewer #2: Yes

5. Is the manuscript presented in an intelligible fashion and written in standard English?

Reviewer #1: Yes

Reviewer #2: Yes

6. Review Comments to the Author

You may also provide optional suggestions and comments to authors that they might find helpful in planning their study.

Reviewer #1: # **Reviewer Comments**

## **Overall Comment**

This protocol addresses an **important and timely topic area**, and it is clear that the authors have put substantial effort into the conceptual framing of patient participation. The manuscript is **clearly written**, well organised, and demonstrates thoughtful consideration of the broad and contemporary conceptualisation of patient participation. A further strength of the protocol is the **use of multiple databases**, which enhances the comprehensiveness of the evidence capture. The topic is relevant, and the protocol has the potential to make a valuable contribution to the literature.

## **Major Comments**

### **1. Search strategy – clarification needed**

Although several databases were included, it is unclear **whether the search strategy was developed with input from a health sciences librarian or information specialist**.

Given best practice for scoping and systematic reviews, please clarify whether librarian input was sought. If not, consider engaging a librarian and documenting their involvement.

### **2. Definition of patient participation – needs alignment with inclusion criteria**

The Introduction adopts a **narrow definition** of patient participation, focusing largely on participation in decision making.

However, your inclusion criteria rely on a **broader, contemporary understanding** of participation—including engagement, empowerment, partnership, and involvement.

These two conceptualisations misalign. I recommend revising the Introduction to reflect the broader definition used in the Methods. The decision‑making‑centric conceptualisation is now quite outdated and grounded in a traditional medical model; contemporary frameworks describe participation as **multidimensional and wide‑ranging**.

### **3. Thematic narrative analysis – require methodological clarity**

Although you state that the review will use **thematic narrative analysis**, the protocol does not specify:

* which methodological framework or reference underpins this analysis, and

* the detailed steps you will follow.

Please add a citation for the methodological approach and outline your analytic process clearly. As this is a protocol, transparency is essential so readers can later evaluate the consistency between the planned and completed review.

### **4. Research Question #3 – clarify approach to “mapping evidence gaps”**

Research Question #3 relates to identifying gaps in the evidence base, but the process for achieving this is not fully described.

As phrased, this could be mistaken for *evidence gap mapping*, a distinct methodology.

Consider clarifying:

* how you will identify and present gaps within the scoping review framework, and

* whether you will use a structured approach such as the **PAGER framework** (Bradbury-Jones et al., 2022) to support the reporting and interpretation of gaps. This would strengthen coherence with your stated aims.

### **5. Alignment with Levac et al. – restructure for clarity**

Although you refer to Levac et al. (2010), the Methods section does not clearly map onto their six-stage framework:

1. Identifying the research question

2. Identifying relevant studies

3. Selecting studies

4. Charting the data

5. Collating, summarising, and reporting the results

6. Consulting with stakeholders

To improve transparency and methodological alignment, consider:

* **renaming subheadings** to reflect these stages, or

* **explicitly stating** within each section which Levac step is being addressed.

For example:

* *Research Questions* → **Step 1: Identifying the Research Question**

* *Search Strategy* → **Step 2: Identifying Relevant Studies**

Also, **Step 6 (Consultation with Stakeholders)** is currently absent and should be added or justified.

## **Minor Comments**

* If date limits are applied to the search, please briefly justify them.

* Ensure consistent use of terminology (e.g., participation, engagement, involvement) once you revise the conceptual framing.

Reviewer #2: Thank you for your submission.

After review, I found that the manuscript is presented more as a protocol or proposal rather than a complete research paper, as it does not include full findings and discussion. The title and topic are strong and promising, especially for advancing patient‑centered decision making, but the work needs to be developed into a full article before it can be considered for publication.

7. PLOS authors have the option to publish the peer review history of their article (what does this mean?). If published, this will include your full peer review and any attached files.

Reviewer #1: No

Reviewer #2: **Yes:**Ola Mousa

---

## [Author Response · Author response to Decision Letter 1]

28 Mar 2026

I have uploaded a document entitled "Respond to Reviewers" which contains our point-by-point responses to the reviewers' comments.

---

## [Decision Letter · Decision Letter 1]

15 Apr 2026

Evidence on strategies promoting discharge planning participation in patients with chronic heart failure: A scoping review protocol

PONE-D-25-67198R1

Dear Dr. Shouqing,

We’re pleased to inform you that your manuscript has been judged scientifically suitable for publication and will be formally accepted for publication once it meets all outstanding technical requirements.

Kind regards,

Divya Bhandari

Academic Editor

PLOS One

Additional Editor Comments (optional):

Thank you for submitting the revised manuscript. The revisions have been carefully evaluated alongside the reviewer’s comments and your detailed responses. The revised manuscript has improved in clarity of conceptual framing, methodological rigor and transparency, and alignment with current scoping review standards and frameworks. Based on the positive reviewer feedback and the satisfactory revision, I am pleased to inform you that your protocol manuscript is accepted for publication in PLOS ONE.

Reviewers' comments:

Reviewer's Responses to Questions

**Comments to the Author**

1. Does the manuscript provide a valid rationale for the proposed study, with clearly identified and justified research questions?

Reviewer #1: Yes

2. Is the protocol technically sound and planned in a manner that will lead to a meaningful outcome and allow testing the stated hypotheses?

Reviewer #1: Yes

3. Is the methodology feasible and described in sufficient detail to allow the work to be replicable?

Reviewer #1: Yes

4. Have the authors described where all data underlying the findings will be made available when the study is complete?

Reviewer #1: Yes

5. Is the manuscript presented in an intelligible fashion and written in standard English?

Reviewer #1: Yes

6. Review Comments to the Author

You may also provide optional suggestions and comments to authors that they might find helpful in planning their study.

Reviewer #1: Well done on responding to feedback. The review team has provided good response to the feedback raised, and the manuscript has improved in quality.

7. PLOS authors have the option to publish the peer review history of their article (what does this mean?). If published, this will include your full peer review and any attached files.

Reviewer #1: No

---

## [Editor Report · Acceptance letter]

PONE-D-25-67198R1

PLOS One

Dear Dr. Shouqing,

I'm pleased to inform you that your manuscript has been deemed suitable for publication in PLOS One. Congratulations! Your manuscript is now being handed over to our production team.

Kind regards,

on behalf of

Ms. Divya Bhandari

Academic Editor

PLOS One